# TOWARDS DISENTANGLING NON-ROBUST AND ROBUST COMPONENTS IN PERFORMANCE METRIC

## ABSTRACT

The vulnerability to slight input perturbations is a worrying yet intriguing property of deep neural networks (DNNs). Though some efforts have been devoted to investigating the reason behind such adversarial behavior, the relation between standard accuracy and adversarial behavior of DNNs is still little understood. In this work, we reveal such relation by first introducing a metric characterizing the standard performance of DNNs. Then we theoretically show this metric can be disentangled into an information-theoretic non-robust component that is related to adversarial behavior, and a robust component. Then, we show by experiments that DNNs under standard training rely heavily on optimizing the non-robust component in achieving decent performance. We also demonstrate current state-of-the-art adversarial training algorithms indeed try to robustify DNNs by preventing them from using the non-robust component to distinguish samples from different categories. Based on our findings, we take a step forward and point out the possible direction of simultaneously achieving decent standard generalization and adversarial robustness. It is hoped that our theory can further inspire the community to make more interesting discoveries about the relation between standard accuracy and adversarial robustness of DNNs.

## 1 INTRODUCTION

Deep neural networks (DNNs) have achieved wide success over the last decade. In literature, the majority of deep learning models pursue boosted performance from different aspects (Krizhevsky et al., 2012; Simonyan & Zisserman, 2014; Szegedy et al., 2015; He et al., 2016; 2015; Ioffe & Szegedy, 2015). However, it is found these powerful models are susceptible to perturbations, even those imperceptible to humans (Szegedy et al., 2013). Therefore beyond the main research stream, there are also works devoted to investigating effective attacks (Goodfellow et al., 2014; Moosavi-Dezfooli et al., 2016; Kurakin et al., 2016; Zhao et al., 2018) and designing adversarial robust models (Goodfellow et al., 2014; Miyato et al., 2015; Madry et al., 2017; Zhang et al., 2019).

It is observed that the adversarial robustness is usually achieved at the cost of a non-trivial degradation of standard performance. Previous efforts (Schmidt et al., 2018; Tsipras et al., 2018; Nakkiran, 2019) trying to understand this phenomenon are usually based on simple toy models or heavy assumptions, and do not provide a general theoretic framework that explicitly shows how adversarial robustness is related to standard generalization. The work (Ilyas et al., 2019) empirically shows the adversarial samples might be human imperceptible features that could help generalization, but it does not provide any general theoretic framework to properly explain this phenomenon. In another work (Zhang et al., 2019), though a bound on the gap between standard accuracy and adversarial accuracy is derived so that the gap between these two quantities could be explicitly controlled, the underlying reason of such trade-off is still not fully characterized.

In this work, we start from a new perspective and consider the performance of deep classification models with a new metric *Cross Category Kullback-Leibler divergence (CCKL)*, namely the Kullback–Leibler (KL) divergence between the model's output distributions over input data from different categories, instead of the traditional metric of accuracy. Interestingly, by applying Taylor expansion on CCKL, we show that it can be disentangled into a lower order non-robust component, which is related to causing adversarial behavior, and a higher order robust component. By applying such

disentanglement, we are able to reveal the relation between standard generalization and adversarial robustness of the deep learning model in a relatively general setting.

Furthermore, we demonstrate by experiments that current deep learning models rely heavily on optimizing the lower order non-robust component to generalize, which is a major underlying reason for the adversarial behavior. We also show the state-of-the-art adversarial training algorithms are all in fact trying to constrain the model from using the lower order non-robust component to discriminate data of different categories. Based on these findings, we claim that enabling the model to rely more on the higher order robust component instead of the adversary-prone lower order component might be the key to achieving decent standard accuracy and adversarial robustness simultaneously.

Our contributions are summarized as follows:

- We propose a new metric, *Cross Category Kullback-Leibler divergence (CCKL)*, to characterize the standard performance. This metric can be more naturally connected with the adversarial behavior of DNNs than the traditional metric of accuracy.

- By applying a simple Taylor expansion on CCKL, we theoretically reveal the relation between standard generalization and adversarial behavior of DNNs in a general way without relying on toy models.

- Based on the above novelties, we take a further step and point out the possible direction for simultaneously achieving decent standard accuracy and adversarial robustness.

## 2 RELATED WORK

**Adversarial Attack and Defense**  To study the adversarial behavior of DNNs, gradient-based algorithms (Goodfellow et al., 2014; Papernot et al., 2016a; Moosavi-Dezfooli et al., 2016; Carlini & Wagner, 2017; Kurakin et al., 2016; Zhao et al., 2018) are developed to find the most effective adversarial perturbations, and on the other hand, model-based defense algorithms (Papernot et al., 2016b; Madry et al., 2017; Miyato et al., 2015; Zhang et al., 2019) are also explored for enhancing model robustness against the adversarial perturbations. For example, in (Miyato et al., 2015; Zhang et al., 2019), the model is regularized by the KL divergence between outputs of clean and adversarial samples, which is also adopted in our theoretic framework.

**Fisher Information in Deep Learning**  Fisher information is used as a tool to study machine learning topics. Many previous works (Chaudhry et al., 2018; Pascanu & Bengio, 2013; Desjardins et al., 2015; Andrychowicz et al., 2016; Achille et al., 2019) view the parameters of the model as those of the Fisher information, while some other (Miyato et al., 2015; Zhao et al., 2018) take the model's input as the parameters of Fisher information, making it applicable in the context of adversarial robustness. We adopt the same definition for Fisher information as (Miyato et al., 2015; Zhao et al., 2018) in our later analysis.

**Explaining Adversarial Behavior of DNNs**  Previous works trying to explain the robust model's degradation in standard performance (Tsipras et al., 2018; Schmidt et al., 2018; Nakkiran, 2019) are mostly based on some simple toy models that cannot generalize to real world settings. Recently, (Ilyas et al., 2019) empirically shows that adversarial noise is actually features that can help the generalization of DNNs. However, this work does not provide a general theoretic framework to explain this phenomenon. Zhang et al. (2019) provides a bound that characterizes the gap between standard accuracy and adversarial robustness, but they do not clearly explain the underlying reason of such trade-off. In our work, we analyze the underlying reason for the adversarial behavior of DNNs by directly disentangling the proposed CCKL. In this way, we not only theoretically demonstrate the reasoning behind the trade-off between adversarial robustness and standard generalization, but also point out the possible direction for achieving both objectives simultaneously.

## 3 NEW PERFORMANCE METRIC AND ITS DISENTANGLEMENT

In this section, we first propose to consider and measure the performance of deep classification models from the perspective of CCKL instead of accuracy. Then we explain our finding that the performance

of a DNN model is determined by two components that are disentangled from the obtained objective based on KL divergence, one of which related to the model's adversarial behavior and the other not. After that, to help understand our idea, we provide some complementary understandings from the viewpoint of information geometry.

## 3.1 PROPOSED PERFORMANCE METRIC

Normally, DNNs are trained by minimizing the following cross-entropy loss:

$$L = \mathbb{E}_{(x,y)\sim\mathcal{X}\times\mathcal{Y}}[KL(y\|f(x))], \tag{1}$$

where $\mathcal{X}\times\mathcal{Y}$ is the distribution where the data-label pair $(x,y)$ is sampled from and $f(x)$ is the prediction of the model. Note that $y$ and $f(x)$ are both discrete distribution over categories here. Correspondingly, the prediction accuracy is adopted to measure the performance of a DNN model. However, such a metric hides the connection between the model performance and its adversarial behavior. Since adversarial attack is concerned with changing the prediction of the model on a data point to another, we propose to shift the performance metric from *investigating one single data point* to *comparing the data point with its counterparts from other categories*, which can help reveal the above connection. Therefore, we propose the following *Cross Category KL Divergence* (CCKL) metric:

$$CCKL = \mathbb{E}_{((x_i,y_i),(x_j,y_j))\in\mathcal{S}}[KL(f(x_i)\|f(x_j))], \tag{2}$$

where $\mathcal{S}$ is defined as

$$\mathcal{S} = \{((x_i,y_i),(x_j,y_j)) \mid \forall i,j \in \mathcal{N}, y_i \neq y_j, x_i, x_j \in \mathcal{X}\}. \tag{3}$$

With the definition above, we can derive the following theorem:

**Theorem 3.1** *Denoting $JS(p\|q)$ as the Jensen-Shannon divergence between two distributions $p$ and $q$, we have*

$$CCKL \geq \mathbb{E}_{((x_i,y_i),(x_j,y_j))\in\mathcal{S}}[JS(y_i\|y_j)] - L - coL, \tag{4}$$

*where $L$ is the cross-entropy loss defined in Equation (1) and $coL = \mathbb{E}_{(x,y)\sim(\mathcal{X},\mathcal{Y})}[KL(f(x)\|y)]$ is the co-object for $L$.*

We defer the proof and more comments of Theorem 3.1 to Appendix A due to space limit.

Taking a closer look at Theorem 3.1, we know that $\mathbb{E}_{((x_i,y_i),(x_j,y_j))\in\mathcal{S}}[JS(y_i\|y_j)]$ remains unchanged during training, and $L$ and $coL$ are supposed to decrease when the model performance increases. Therefore, the lower bound on CCKL is supposed to increase with the rise of model performance, which will asymptotically result in the increase of CCKL. Thus, the CCKL, which characterizes how well the model discriminates input data from different categories, can serve as a performance metric.

To further expound on our point, we provide a visualization of how CCKL evolves with test accuracy and test cross entropy loss during training in Figure 2. According to our results, the variation of CCKL correlates well with the test accuracy and test cross entropy loss during training.

## 3.2 CONNECTIONS BETWEEN ADVERSARIAL BEHAVIOR AND CCKL

We now explain how the above CCKL objective (2) correlates with the model's adversarial behavior. Consider a single data-label pair $(x,y)$. The following cross entropy loss of the model $f_\theta$ on adversarial samples (with perturbation $\eta$):

$$L_\theta(x,\eta) = KL(y\|f_\theta(x+\eta)) \text{ s.t. } \|\eta\| \leq \epsilon \tag{5}$$

is adopted to study the adversarial behavior in most previous literature (Goodfellow et al., 2014; Kurakin et al., 2016; Madry et al., 2017). However, it is not easy to build connection between this objective and the standard generalization of the model. Interestingly, in some recent works (Miyato et al., 2015; Zhao et al., 2018; Zhang et al., 2019), another objective is adopted to characterize the model's adversarial behavior:

$$L_\theta(x,\eta) = KL(f_\theta(x)\|f_\theta(x+\eta)) \text{ s.t. } \|\eta\| \leq \epsilon. \tag{6}$$

Based on the above objective (6), (Zhao et al., 2018) reports state-of-the-art results in the task of adversarial attack and (Zhang et al., 2019) reports state-of-the-art results in adversarial robustness.

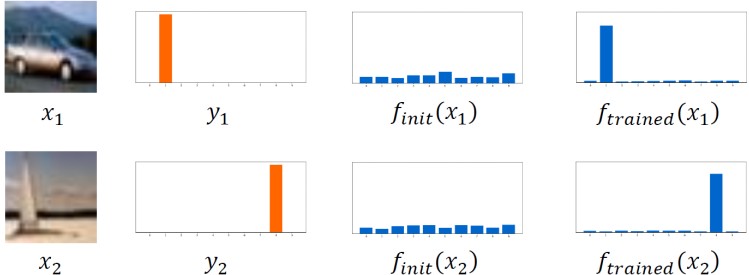

Figure 1: Illustration on how output distributions $f(x_1)$ and $f(x_2)$ evolve during training. Column 2: ground-truth label. Column 3: network outputs at initialization. Column 4: network outputs when trained to converge. The value of $KL(f_{init}(x_1)\|f_{init}(x_2))$ in the graph is 0.053, and the value of $KL(f_{trained}(x_1)\|f_{trained}(x_2))$ is 3.197.

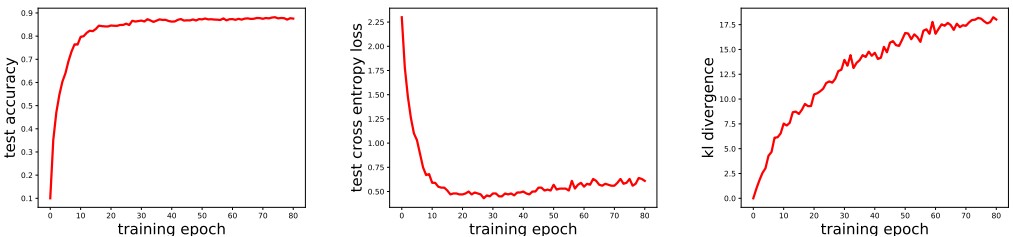

Figure 2: Visualization of how accuracy (left), cross entropy loss (middle) and CCKL (right) on test set evolve when training VGG13 on CIFAR-10. It is clear that CCKL well indicates test accuracy and test cross entropy loss. For more experiment results on relation between CCKL and cross entropy loss, please refer to Appendix B.

More importantly, as will be shown later in the paper, the adversarial objective (6) can be easily linked to the standard generalization of the model. Therefore, we adopt the objective (6) in the rest of the paper. The corresponding adversarial training objective is thus formulated as follows:

$$\min_{\theta} \max_{\eta} L_{\theta}(x, \eta) \text{ s.t. } \|\eta\| \leq \epsilon. \tag{7}$$

Given the definition of $L_{\theta}(x, \eta)$, applying Taylor expansion yields the following:

$$\max_{\eta} L_{\theta}(x, \eta) = \max_{\eta} \frac{1}{2}\eta^{\top} F_x \eta + \sum_{k=3}^{\infty} T_x^{(k)}(\eta) \text{ s.t. } \|\eta\| \leq \epsilon, \tag{8}$$

where $F_x$ is the Fisher information of $f(x)$ w.r.t. $x$ and $T_x^{(k)}$ is the following $k$-th order Taylor expansion term. Let $f^{(j)}(x)$ be the $j$-th entry of $f(x)$ and $n$ be the dimension of $f(x)$. Then $F_x$ can be calculated by

$$F_x = \sum_{j=0}^{n} f^{(j)}(x)(\nabla_x \log f^{(j)}(x))(\nabla_x \log f^{(j)}(x))^{\top}. \tag{9}$$

When $\epsilon$ is sufficiently small, the higher order terms in the above would vanish and Equation (8) could be simplified into

$$\max_{\eta} L_{\theta}(x, \eta) = \frac{1}{2} \max_{\eta} \eta^{\top} F_x \eta \text{ s.t. } \|\eta\| \leq \epsilon. \tag{10}$$

We could therefore construct the lagrangian of the above problem:

$$L(x, \eta) = \frac{1}{2}\eta^{\top} F_x \eta - \frac{1}{2}\lambda(\eta^{\top}\eta - \epsilon^2). \tag{11}$$

By setting $\nabla_{\eta} L(x, \eta) = 0$, we obtain $F_x \eta = \lambda_{\max} \eta$, where $\lambda_{\max}$ is the maximum eigenvalue of $F_x$. Therefore, the most effective adversarial perturbation corresponds to the leading eigenvector of $F_x$.

Consequently, we have

$$\max_{\eta} L_\theta(x, \eta) = \frac{1}{2}\lambda_{\max}\epsilon^2. \tag{12}$$

Note that $\lambda_{\max}$ here is also the spectral norm of the Fisher information matrix $F_x$. The above theory shows that the local adversarial behavior of the model $f$ around the input $x$ is determined by the spectral norm of the Fisher information matrix: the adversarial behavior around $x$ would be more severe if the spectral norm of $F_x$ is larger.

Now we analyze CCKL by considering one tuple $((x_i, y_i), (x_j, y_j))$ randomly sampled from $\mathcal{S}$. We rewrite $KL(f(x_i)\|f(x_j))$ as

$$KL(f(x_i)\|f(x_j)) = KL(f(x_i)\|f(x_i + (x_j - x_i))) = L_\theta(x_i, x_j - x_i). \tag{13}$$

Then, we apply the same Taylor expansion as above and obtain

$$KL(f(x_i)\|f(x_j)) = L_\theta(x_i, x_j - x_i) = \frac{1}{2}(x_j - x_i)^\top F_{x_i}(x_j - x_i) + \sum_{k=3}^{\infty} T_{x_i}^{(k)}(x_j - x_i). \tag{14}$$

Comparing Equation (14)[1] and Equation (10), we notice they share the same Fisher information $F_{x_i}$ at data point $x_i$. Here, we show the adversarial behavior at each data point can be connected to the performance objective CCKL via Fisher information, which lays the foundation for the disentanglement in the next subsection.

### 3.3 DISENTANGLEMENT OF CCKL

As mentioned above, we build the connection between adversarial behavior and CCKL. We here explain how the non-robust and robust component in CCKL can be disentangled. We first denote $\frac{1}{2}(x_j - x_i)^\top F_{x_i}(x_j - x_i)$ in Equation (14) as $G_1$ and the following terms $\sum_{k=3}^{\infty} T_{x_i}^{(k)}(x_j - x_i)$ as $G_2$. Thus $KL(f(x_i)\|f(x_j))$ can be formulated as

$$KL(f(x_i)\|f(x_j)) = G_1 + G_2. \tag{15}$$

Taking a closer look at Equation (15), we notice the increase of $G_1$ and $G_2$ can both contribute to the rise of $KL(f(x_i)\|f(x_j))$, which contribute to CCKL. In addition, since $\lambda_{max}$ is the maximum eigen value of $F_{x_i}$, we have the following bound:

$$\frac{1}{2}\lambda_{\max}\|\eta\|^2 \geq \frac{1}{2}\left[\frac{\|\eta\|}{\|x_i - x_j\|}(x_i - x_j)\right]^\top F_{x_i}\left[\frac{\|\eta\|}{\|x_i - x_j\|}(x_i - x_j)\right] = G_1\frac{\|\eta\|^2}{\|x_i - x_j\|^2}. \tag{16}$$

With the above bound, we know that the rise of $G_1$ would asymptotically result in the rise of the norm of $F_{x_i}$, which corresponds to the adversarial behavior.

Therefore, if the model relies heavily on the increase of $G_1$ to boost performance, the norm of $F_{x_i}$ would also increase drastically. According to Equation (12) and our analysis above, the rise in the norm of $F_{x_i}$ means more severe adversarial behavior around $x_i$. The trade-off between standard performance and adversarial behavior is thus characterized as follows: the model can rely on $G_1$ to boost performance, but it comes with the side effect of more severe adversarial behavior; on the other hand, $G_2$ contributes to the CCKL but it is not involved in the adversarial objective, thus it would not cause adversarial behavior to rely on terms in $G_2$ to distinguish $x_i$ from data belonging to other categories. In this way, we can disentangle the non-robust and robust component in the overall performance objective CCKL.

To further understand the role of $G_1$ in classification, we visualize how F-norm of Fisher information evolves during training. We visualize F-norm instead of spectral norm because all norms are equivalent and spectral norm is not computationally feasible in our case. We first empirically show how the average F-norm of Fisher information and standard accuracy of VGG13 model on CIFAR-10 test set vary during normal training in Figure 4. We observe that the norm of Fisher information increases drastically with the rise of accuracy, which indicates DNNs rely heavily on the non-robust component $G_1$ to boost performance.

---

[1]Note that $(x_j - x_i)$ is the difference between two input data instead of a small perturbation, so the higher order terms would not vanish here.

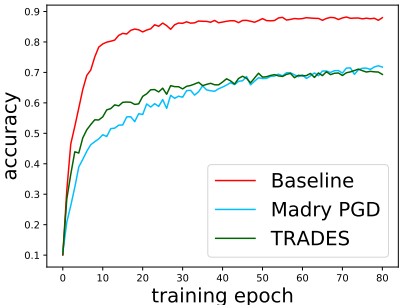 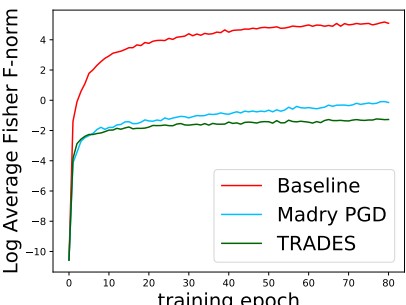

Figure 3: Visualization of how standard test accuracy (left) and average F-norm (right) of Fisher Information Matrix on test set vary during normal training and adversarial training with VGG13 on CIFAR-10 (for the same experiments on ResNet, see Appendix D). We take the nature logarithm to better visualize the average F-norm of Fisher information.

Then we compare normal training with the two state-of-the-art adversarial training algorithms (Madry et al., 2017; Zhang et al., 2019) using the same visualization method, and show results in Figure 3. It is clear that during adversarial training, although the Fisher information's average F-norm also rises with standard accuracy, its value is significantly smaller than its counterpart during normal training. That is, the adversarial training process can effectively constrain the model from relying on Fisher information to boost performance.

Our experiments reveal the widely known but seldom understood fact that the standard accuracy of models under the two adversarial training algorithms are significantly lower than that under normal training. According to our theory, it can be explained as these models trained over adversarial samples are unable to effectively rely on adversary-prone non-robust components such as Fisher information to distinguish the input data from those belonging to other categories.

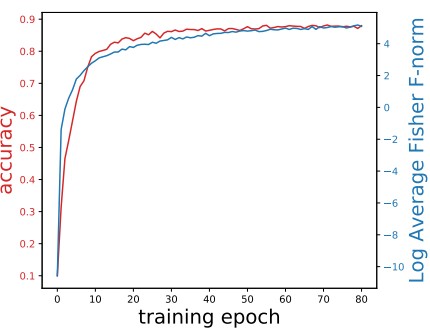

Figure 4: Visualization of how standard accuracy and average F-norm of Fisher information matrix on test set vary during normal training. The experiment is conducted with a VGG13 model and CIFAR-10 data set (for the same experiment on ResNet, see Appendix D). We take the nature logarithm to better visualize the average F-norm of Fisher information.

Till now, we theoretically and empirically demonstrate the relation between standard performance and adversarial robustness using the proposed disentanglement theory.

### 3.4 EXPLANATION FROM GEOMETRY POINT OF VIEW

We provide some explanations on the above findings from the information geometry viewpoint. We note the model is doing maximum likelihood estimation by learning to fit the label distribution over training data. This can be viewed as a process of the log-likelihood landscape of the model on input data gradually transiting into a state where the model can well distinguish data of different categories. Since the training data are sparsely sampled from the whole distribution, the smoothness prior does not hold during the formation of the model's log-likelihood landscape. Consequently, the model tends to use lower order local geometric descriptors such as Fisher information —the local curvature of the log-likelihood landscape —to form an overly simplified adversary-prone log-likelihood landscape. When applying adversarial training, a strong smoothness constraint is enforced and the model would have to rely on higher-order global geometric descriptors that vanish locally to form the whole landscape. Thus the landscape would be more robust to adversarial perturbations.

Table 1: Standard accuracy and adversarial robustness of models of different capacity trained by (Zhang et al., 2019). We set the maximum allowed $L_\infty$ norm of attack noise $\epsilon = 8/255$. The learning rate is 0.01 for all VGG13 models and 0.1 for all resnet models. All models are trained with SGD for 160 epochs with a decay of $10\times$ in learning rate at 80th and 120th epochs. The SGD's momentum is 0.9. For adversarial training settings, the coefficient for the regularization term in (Zhang et al., 2019) is $\frac{1}{\lambda} = 5.0$, the step size for projected gradient descent is 2/255 and the number of step is 10. The weight decay during training is 1e-4. For evaluation against adversarial attack, the step size for PGD attack is 2/255, the number of steps is 20. The CW attack objective's coefficient is $c = 5e2$ and is also solved by PGD with the same optimization parameters. All experiments are conducted on NVIDIA Tesla V100 GPUs.

|  | VGG13 pure linear | VGG13 half linear | VGG13 normal | resnet20x1 | resnet32x10 |
|---|---|---|---|---|---|
| accuracy (standard) | 35.14 | 70.48 | 72.36 | 72.88 | 79.45 |
| accuracy (PGD attack, $\epsilon = 8/255$) | 17.06 | 38.52 | 42.29 | 44.94 | 49.52 |
| accuracy (CW attack, $\epsilon = 8/255$) | 13.94 | 34.12 | 38.96 | 41.49 | 47.67 |

## 3.5 Analysis for $\epsilon$ Not Sufficiently Small

Researchers have found that the local linearity does not hold when the norm of perturbation $\epsilon$ is allowed to be relatively larger (Kurakin et al., 2016; Madry et al., 2017). In this case, some higher order terms may also involve in the adversarial objective. However, since the norm of perturbation is still considered to be small, the Taylor expansion terms of $KL(f(x)\|f(x+\eta))$ would still vanish up to a certain order $K_\epsilon$. Therefore, the adversarial objective can be rewritten as

$$\max_\eta L_\theta(x, \eta) = \sum_{k=2}^{K_\epsilon} T_x^{(k)}(\eta) \ \text{ s.t. } \|\eta\| \leq \epsilon. \tag{17}$$

Comparing (14) and (17), we know that the functional of the first $K_\epsilon - 1$ terms could still jointly connect adversarial objective and CCKL. Also, we conduct numeric experiments in Appendix C to show that the Fisher induced term is the dominant component in $KL(f(x)\|f(x+\eta))$ within a practical range of perturbation. We leave more detailed theoretic analysis on higher order terms involved in the adversarial objective to future work.

## 4 Towards Simultaneous Good Performance and Robustness

With the disentanglement theory introduced above, we now consider how to achieve simultaneous decent standard accuracy and adversarial robustness. According to our analysis, if relying on the robust component alone can effectively distinguish data of different categories, then obtaining an adversarial robust model with high standard accuracy is possible.

Since the expansion terms in the robust component are all higher order terms, we claim that the key to achieving the two desired qualities simultaneously is to increase the expressive power of the model so that it would be able to rely more on higher order terms for prediction. In this way, the model would not have to rely heavily on Fisher information and still have decent standard performance with higher order terms (the robust component). Our disentanglement provides theoretical justification for the importance of model capacity in achieving adversarial robustness and decent standard performance.

We experiment on CIFAR-10 to validate our strategy of achieving the two objectives simultaneously. We train the following models with TRADES algorithm (Zhang et al., 2019): a pure linear VGG13 model without ReLU and with average pooling, a half linear VGG13 model without ReLU but using max pooling, a normal VGG 13 model, a normal resnet20 model and a resnet32 model with $10\times$ more channels. We evaluate these models on standard samples and adversarial samples produced

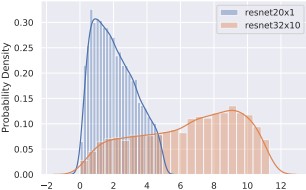 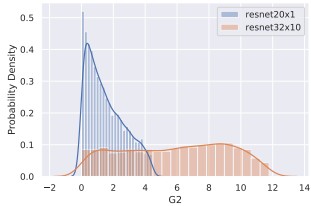 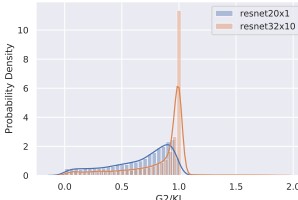

Figure 5: We sampled 25896 pairs of images $(x_i, x_j)$ in CIFAR-10 test set ($x_i$ and $x_j$ belong to different categories) and visualized the distribution of $KL(f(x_i)\|f(x_j))$ (left), distribution of $G_2$ (middle), and distribution of $G_2/KL(f(x_i)\|f(x_j))$ (right) of resnet20x1 and resnet32x10.

by $L_\infty$ PGD attack (Madry et al., 2017) and CW attack (Carlini & Wagner, 2017). The results are provided in Table 1.

We first compare VGG models. Since adversarial training impedes the model from using the adversary-prone lower order terms to discriminate input data, and the pure linear model is not capable of utilizing higher order information, the standard generalization and adversarial robustness of the pure linear model are both very poor. However, as the capacity of the model increases (half linear and normal VGG13 models), the performance on standard and adversarial samples improves simultaneously. For resnet models, when the model is shallower and narrower (resnet20x1), the performance in both standard and adversarial settings is relatively low. However, with a deeper and wider resnet model (resnet32x10), more higher order information is explored for prediction, so the performance on standard samples and adversarial samples increases significantly.

In addition, we sampled 25896 pairs of images $(x_i, x_j)$ in CIFAR-10 test set ($x_i$ and $x_j$ belong to different categories) and visualized the distribution of $KL(f(x_i)\|f(x_j))$, $G_2$, and $G_2/KL(f(x_i)\|f(x_j))$ with the above mentioned adversarially trained resnet20x1 and resnet32x10 model in Figure 5.

According to our results in Figure 5, we have the following observations. 1) The value of CCKL of the resnet32x10 model on CIFAR-10 test set is much higher than that of resnet20x1, which means the resnet32x10 can better distinguish input data from different categories. 2) The value of $G_2$ of resnet32x10 on the CIFAR-10 test set is significantly higher than that of resnet20x1, which shows $G_2$ plays a much more important role in resnet32x10 than resnet20x1. 3) The ratio of $G_2/KL(f(x_i)\|f(x_j))$ of resnet32x10 on CIFAR-10 test set is much higher than that of resnet20x1, meaning the relative importance of $G_2$ of resnet32x10 is much greater than that of resnet20x1.

These observations validate our claim that a model with better capacity can more effectively leverage higher order information to achieve higher standard accuracy and also adversarial robustness under adversarial training.

## 5 CONCLUSION

In this work, we provide a novel viewpoint on the relation of standard accuracy and adversarial robustness of deep learning models. We propose a new CCKL metric to measure the model performance instead of accuracy. With CCKL, the overall performance objective can be disentangled into an adversary-prone non-robust component, and a robust component. Based on such disentanglement, we then claim that for achieving both adversarial robustness and decent standard accuracy, a DNN model should rely more on the robust component to generalize. Our findings are well validated. In the future, we will dig deeper along the geometric properties of the log-likelihood landscape formed by DNNs on input data, and try to better characterize the relation between standard accuracy and adversarial robustness.

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

## A    PROOF AND COMMENTS OF THEOREM 3.1

In practical scenario, one cannot be too confident about a label $y$ due to human annotation errors, and often replaces it during training with smoothed version $y^{LS}$ using the label smoothing technique Szegedy et al. (2016). Namely,

$$y^{LS} = \alpha y + (1 - \alpha)\frac{1}{K},$$

where $K$ is the number of classes, and $\alpha \in [0, 1]$ is the confidence level. The label smoothing version of Theorem 3.1 still holds.

Proof of Theorem 3.1:

$\forall$ data-label pair $(x_i, y_i)$, $(x_j, y_j)$, where $y_i \neq y_j$, since $JS^{1/2}$ satisfies triangular inequality (Endres & Schindelin, 2003), we have:

$$
\begin{aligned}
& JS(f(x_i)\|f(x_j))^{1/2} + JS(y_i\|f(x_i))^{1/2} + JS(y_j\|f(x_j))^{1/2} \\
& \geq JS(y_i\|f(x_j))^{1/2} + JS(y_j\|f(x_j))^{1/2} \\
& \geq JS(y_i\|y_j)^{1/2}
\end{aligned}
\tag{18}
$$

Also, we have the inequality:

$$(a + b + c)^2 \leq 2(a^2 + b^2 + c^2).
\tag{19}$$

With (18) and (19), we have:

$$
\begin{aligned}
& 2(JS(f(x_i)\|f(x_j)) + JS(y_i\|f(x_i)) + JS(y_j\|f(x_j))) \\
& \geq (JS(f(x_i)\|f(x_j))^{1/2} + JS(y_i\|f(x_i))^{1/2} + JS(y_j\|f(x_j))^{1/2})^2 \quad \text{(applying (19))} \\
& \geq JS(y_i\|y_j). \quad \text{(applying (18))}
\end{aligned}
\tag{20}
$$

In addition, we have Lin's inequality (Lin, 1991):

$$JS(p\|q) \leq \frac{1}{4}(KL(p\|q) + KL(q\|p)).
\tag{21}$$

Apply (21) to (20), we have:

$$
\begin{aligned}
& JS(y_i\|y_j) \\
& \leq 2(JS(f(x_i)\|f(x_j)) + JS(y_i\|f(x_i)) + JS(y_j\|f(x_j))) \\
& \leq \frac{1}{2}(KL(f(x_i)\|f(x_j)) + KL(f(x_j)\|f(x_i)) \\
& + KL(y_i\|f(x_i)) + KL(y_j\|f(x_j)) \\
& + KL(f(x_i)\|y_i) + KL(f(x_j)\|y_j))
\end{aligned}
\tag{22}
$$

Taking the expectation over the data set in (22), we have:

$$CCKL \geq \mathbb{E}_{\forall y_i \neq y_j}[JS(y_i\|y_j)] - \mathbb{E}[KL(y\|f(x))] - \mathbb{E}[KL(f(x)\|y)],
\tag{23}$$

which proves the theorem 3.1.

## B    ADDITIONAL RESULTS ON CCKL AS PERFORMANCE METRIC

We further empirically validate the relation between CCKL and standard cross entropy loss on test set in Figure. 6. We trained a ResNet-20 on CIFAR-10 and CIFAR-100. The Pearson Correlation Coefficients of CCKL and cross entropy loss on test set across the whole training period are **-0.6744** and **-0.8018** in these two settings, which further justify our use of CCKL as performance metric.

## C    APPROXIMATE KL DIVERGENCE LOCALLY WITH FISHER

Experiment in Tab. 2 shows the relative numerical error of approximating KL locally with Fisher induced term with a VGG13 model adversarially trained by (Zhang et al., 2019) on CIFAR-10. We can observe that when $\epsilon$ is within practical range ($\leq 8/255$), the relative numeric error is small.

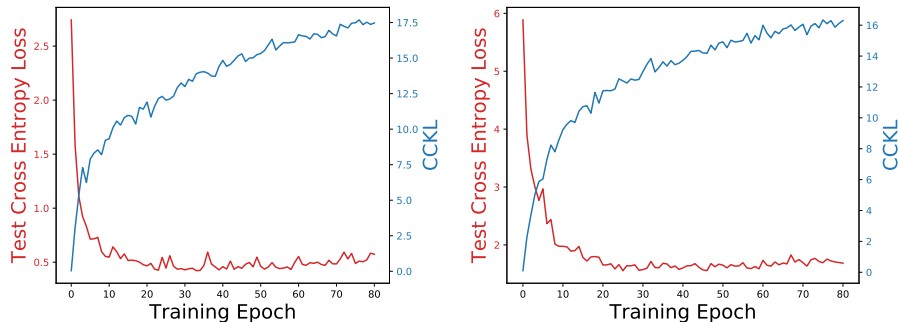

Figure 6: relation between CCKL and test loss of resnet20 on CIFAR-10 and CIFAR-100

Table 2: $\epsilon$ and $G_1$ (We denote $G_1 = \frac{1}{2}\eta^T F_x \eta$ here.)

| $\epsilon$ | 1/255 | 2/255 | 3/255 | 4/255 | 5/255 | 6/255 | 7/255 | 8/255 |
|---|---|---|---|---|---|---|---|---|
| $|KL - G_1|/KL$ | 0.058 | 0.098 | 0.135 | 0.170 | 0.202 | 0.234 | 0.263 | 0.290 |

## D    MORE EXPERIMENTS ON THE ROLE OF FISHER INFORMATION

We conduct more visualization experiments about the role of Fisher information in standard perfor-mance of DNN. The results are shown in Figure 7 and Figure 8. The experiments are conducted on a resnet20 model. The same conclusion could be drawn according to our statistics.

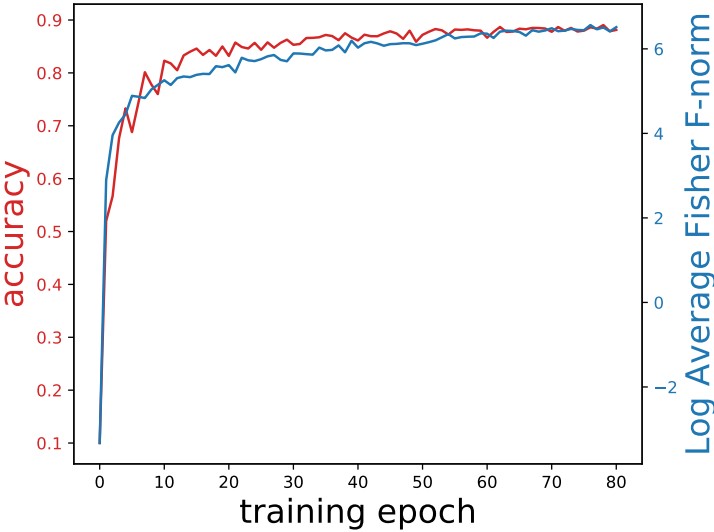

Figure 7: Visualization of how standard accuracy and average F-norm of Fisher information matrix on test set vary during training. The experiment is conducted on a resnet20 model and CIFAR-10 data set. We take the nature logarithm to better visualize the average F-norm of Fisher information

Also, to better understand the geometry of an adversarially trained model around robust samples and non-robust samples, we visualized the F-norm of Fisher information at correct and robust samples and correct but not robust samples of a VGG13 model. According to our results in Figure 9, we could see that although the overall Fisher information is small, the Fisher information around robust samples are still significantly smaller than that of non-robust samples.

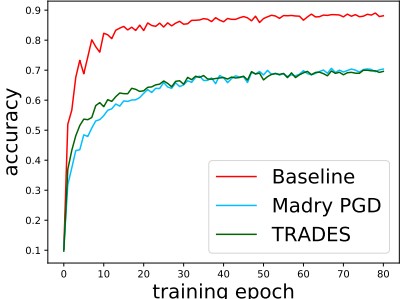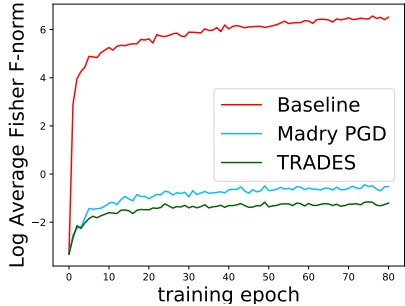

Figure 8: Visualization of how (left) standard test accuracy and (right) average F-norm of Fisher Information Matrix on test set vary during normal training and adversarial training, with resnet20 on CIFAR-10. We take the nature logarithm to better visualize the average F-norm of Fisher information

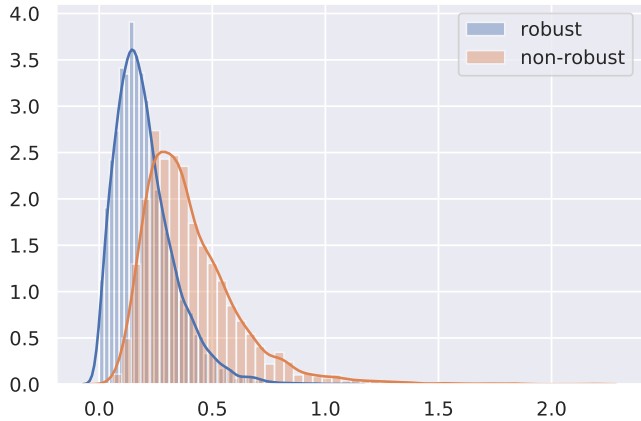

Figure 9: Distribution of F-norm of Fisher information at robust and non-robust samples on CIFAR-10 test set.

## E    INTERPRETATION FROM CRAMÉR-RAO BOUND POINT OF VIEW

According to the main text, adversarial training constrains the input-output Fisher information of a DNN model. This constrain is a criteria of a good DNN model due to the following reasons. Recall the well-known Cramér-Rao bound

$$\mathrm{var}(\hat{x})F_x \geq 1$$

says that if we try to use the output probability $f(x)$ to a statistics $\hat{x}$ to reconstruct the input $x$, the uncertainty in terms of variance $\mathrm{var}(\hat{x})$ is bounded below by the inverse of Fisher information $F_x$. For a DNN model that represents the reality, when it classifies an image with a correct label, say a dog, the label does not have any information about the environments - what color the dog is, where is the dog, adversarial perturbation, etc. Therefore, one cannot use the information contained in the label to reconstruct the original image. This means that the variance $var(\hat{x})$ of any statistics $\hat{x}$ derived from output distribution $f(x)$ is relatively large for a good DNN model. In view of Cramer-Rao bound, this implies that the Fisher information of a DNN is a relatively small value.

