# OpenReview forum: "Towards Disentangling Non-Robust and Robust Components in Performance Metric"
_ICLR.cc/2020/Conference — Reject_

### Official Review · AnonReviewer2 · 2019-10-21
**Official Blind Review #2**

**Rating:** 1

**Review:**

The authors propose a new performance metric, CCKL, as an alternative to standard cross-entropy. At a high level the goal of this objective is to ensure that the classifier is confidently separating samples from different classes. They decompose this objective into its Taylor expansion terms and argue that, since adversarial vulnerability is a local property, it only depends on the first of these terms. They finally propose training models of higher capacity as a way of relying more on higher-order terms of the objective and hence improving adversarial robustness.

I found the core idea potentially interesting. It is an attempt at rigorously studying the tension between local stability (necessary for robustness) and the fact that different classes should be confidently separated. However, I have several concerns regarding the theoretical derivation and experimental evaluation.

First of all, I did not find the proposed CCKL objective well motivated. Theorem 3.1 only shows that standard training improves a _lower bound_ on the proposed objective. However, this only implies that this objective is being optimized as a byproduct of standard cross-entropy optimization. It is unclear for instance if this is a good performance indicator or whether training with this objective will even lead to a good classifier (for instance the classifier could learn a permutation of the labels instead). Therefore, I do not find the theoretical and experimental evidence in favor of studying this proposed objective convincing.

Moreover, the experimental evidence presented is quite weak. While the authors compare various quantities related to standard and robust models, they do not perform the ablations necessary to understand whether these properties are causally related to robustness of simply a byproduct of the specific training scheme.

Finally, the conclusion that larger capacity can improve both model accuracy and robustness has already been made in prior work (Kurakin et al. 2016, Madry et al. 2017). The intuition that added capacity helps by allowing the model to leverage higher order terms of the loss is interesting but a rather incremental contribution.

Overall, while the ideas presented in the paper could be of interest, they are not rigorously established and rather serve as high-level intuition at this moment. Moreover, the writing of the paper could be significantly improved. It took me quite some time to distill the above intuition despite the fact that it is a rather elementary consequence of the Taylor expansion (I am not sure what the discussion about information geometry adds to this work). I would thus encourage the authors to continue working on these ideas and their presentation. However, for now, I will have to recommend rejection.

**Experience Assessment:**

I have published in this field for several years.

**Review Assessment: Checking Correctness Of Derivations And Theory:**

I carefully checked the derivations and theory.

**Review Assessment: Checking Correctness Of Experiments:**

I carefully checked the experiments.

**Review Assessment: Thoroughness In Paper Reading:**

I read the paper thoroughly.

---

> ### Author Response · Authors · 2019-11-11
> **Response to Official Blind Review #2**
>
> Thanks for your efforts in reviewing our work. We hope our following response could help resolve your concerns.
>
> First of all, we never meant to optimize our model based on CCKL. CCKL only need to serves as an indicator of the performance. The overall flow of our idea goes like this:
>
> decrease of cross-entropy loss ==> rise of CCKL ==> rise of non-robust component ==> causing adversarial behavior.
>
> Therefore, as long as it's a byproduct and good indicator of the optimization of cross entropy loss, it would be enough to lay the foundation for our following derivations. We only need "rise of CCKL" to be the necessary condition of "the decrease of cross-entropy loss". As for whether it's a good indicator on the performance, we've already provided a theoretic bound and several experiments in the main text and appendix to justify its role as a good performance indicator to cross-entropy loss.
>
> In addition, as far as we are concerned, we've provided enough experimental evidence in the paper to support our theoretical findings. Our previous theories show that the overall performance metric could be disentangled into a non-robust and a robust component. Next, we show that the non-robust component will rise drastically with the standard accuracy, which shall cause serious adversarial behavior. Also, to prove our point from another direction, we show that adding adversarial training will also greatly impede the rise of the non-robust component. In this way, we also present some strong empirical evidence on the disentanglement in addition to the theoretical derivations. If you think more ablation should be added, please be more specific about it.
>
> We would like to thank you for acknowledging our contribution on explaining the effectiveness of capacity in achieving simultaneous decent performance and adversarial robustness. However, we have to restate here that previous efforts only provide some empirical results and intuitions on this phenomenon, however, we're the first to provide a theoretic justification and understanding on the effectiveness of capacity/non-linearity in achieving simultaneous adversarially robust and decent standard performance. Also, to the best of  our knowledge, our experiments on the non-linearity (Linear VGG/Half-Linear VGG/Standard VGG) is not included in any previous work.
>
> Also, we think although the final step is simply a Taylor expansion, all the previous steps (such as justification on using CCKL as performance metric, justification on connection between Fisher information and adversarial non-robustness) are necessary to rigorously develop the whole idea. As for the information geometry part, since all our previous derivations are based on the log-likelihood landscape, it is natural to introduce an overall explanation on the difference between the formation of the log-likelihood landscape of standard model and robust model. This part was indeed meant to serve as a high level intuition over the whole work and the previous parts are rigorous derivations. However, we only used a small subsection on this part and it doesn't affect the overall flow of the development of our idea.

---

> > ### Comment · AnonReviewer2 · 2019-11-15
> > **Response**
> >
> > Thank you for your response.
> >
> > -- It is now clear that CCKL only serves as an auxiliary loss for your analysis. As such I will not consider it is as a contribution of its own and only evaluate in the context of the other contributions.
> >
> > -- As mentioned in my original review, the experiments performed only establish that adversarial training reduces the contribution of second order terms. In order to claim that this is where robustness comes from, one would need to train with second order regularization and show that this indeed improves model robustness.
> >
> > -- I agree that your work aims to theoretically explain why capacity improves standard and adversarial performance. However, the _empirical_ contributions on that front are minor given prior work.
> >
> > -- I still do not find the additional discussion and derivation particularly insightful. From my point of view, the contribution of the paper can be fully summarized along the lines of studying the Taylor expansion of the models loss. The authors would need to explain what theoretical and empirical insights can only be gained within this framework. I believe that my review summarizes the paper reasonably well while only discussion the contribution of different loss terms.

---

### Official Review · AnonReviewer3 · 2019-10-24
**Official Blind Review #3**

**Rating:** 1

**Review:**

The paper is built around the theme of adversarial attacks. The papers titled "Towards..." are usually records of failed attempts of an attack on a grand challenge. The reviewer generally believes, that such records can be of a value.
In case of this paper an alternative loss function is proposed, based on information theory. The authors claim and support their claims with graphs that this loss tracks the accuracy much more faithfully than the "standard" cross entropy. A natural question would be therefore - what happens if we use this new measure as a training loss? Would it lead to more adversarially-robust models? Should evidence for that be supported for that the reviewer would be strongly in favour of accepting the paper. In the present shape the reviewer does not see why such measure should be of an interest to community. The impact of "inspiring the community to make interesting discoveries" is in the reviewers opinion not sufficient to justify publication at a venue of such importance as ICLF.

**Experience Assessment:**

I have published one or two papers in this area.

**Review Assessment: Checking Correctness Of Derivations And Theory:**

I assessed the sensibility of the derivations and theory.

**Review Assessment: Checking Correctness Of Experiments:**

I assessed the sensibility of the experiments.

**Review Assessment: Thoroughness In Paper Reading:**

I read the paper at least twice and used my best judgement in assessing the paper.

---

> ### Author Response · Authors · 2019-11-11
> **Response to Official Blind Review #3**
>
> First, we have to point out your misunderstandings.  First, CCKL is NOT meant to be a proposed loss function. We NEVER claimed in any way in our paper that CCKL could track accuracy more faithfully than cross entropy loss. This objective is NEVER meant to be optimized. We're merely justifying that the decrease of the standard cross entropy loss directly result in the rise of CCKL, which lays the foundation for our following derivations on the relation between adversarial robustness and standard performance. Throughout the whole work, CCKL serves as a auxiliary role to develop our main theory. For more explanation on the "auxiliary role", please refer to our reply to R2.
>
> In addition, our major contribution is about revealing the relation between the standard performance and adversarial robustness. Based on the proposed CCKL, we further theoretically show that the rise of CCKL could be result from both rise of non-robust component as well as rise of robust component. If the model relies on the non-robust component to boost performance, it will has the side-effect of adversarial vulnerability. If the model doesn't rely heavily on the non-robust component, it will suffer greatly in standard performance. Also, we conducted various experiments to prove our point, such as showing non-robust component will rise drastically during standard training, showing adversarial training will effectively regularize the non-robust component. What's more, based on our theory, we point out the possible direction of achieving simultaneous adversarial robust and decent standard performance, which is enabling the model to rely more on robust component to distinguish data from different categories. All these points presented above are actually the central points of our work, and we believe we have stated clearly in our main text.

---

### Official Review · AnonReviewer1 · 2019-10-26
**Official Blind Review #1**

**Rating:** 3

**Review:**

The work explores the problem of robustness and adversarial
attacks in NN. In a multiclass prediction setting the idea
is to use a taylor expansion of a loss coined CCKL which
is the KL divergence between predictions for pairs of samples
from different classes.

The papers seems to find a convoluted route to arrive
to something like this: when the Fisher information matrix
has a strong eigenvalue the model is not robust. In other words
it says that if the landscape close to convergence has
valleys, or fast changes, the model is not robust.
This appears quite obvious and related to previous similar
studies.

This statement is then empirically evaluated on CIFAR-10.

The mathematical derivations should be made more rigorous.
For example the paragraph on Cramer-Rao bound is very handwavy.

Typos

-  is found these  ->  is found that these

**Experience Assessment:**

I do not know much about this area.

**Review Assessment: Checking Correctness Of Derivations And Theory:**

I assessed the sensibility of the derivations and theory.

**Review Assessment: Checking Correctness Of Experiments:**

I did not assess the experiments.

**Review Assessment: Thoroughness In Paper Reading:**

I read the paper at least twice and used my best judgement in assessing the paper.

---

> ### Author Response · Authors · 2019-11-11
> **Response to Official Blind Review #1**
>
> Thanks for your efforts in reviewing our work. Unfortunately, we have to point out your misunderstanding in our work. The main idea of our work is NOT focused on the relation between Fisher's eigen value and robustness. This relation is simply one bedrock for our conclusion.
>
> Our main purpose of this work is to reveal the relation between standard performance and adversarial robustness. It has been observed that an adversarial robust DNNs usually don't have a comparable standard performance with standard DNNs. However, there is no previous work rigorously showing the underlying reason behind this phenomenon. In our work, we theoretically reveal this little understood relation by disentangling the non-robust and robust component in a proposed performance metric. We show that the model could rely on both the non-robust component as well as robust component to boost performance. And if the model relies heavily on the non-robust component to distinguish data from different categories, it will enjoy a high standard performance, but will also be adversarial vulnerable at the same time.
>
> In addition, we conducted various experiments to prove our point, such as showing non-robust component will rise drastically during standard training, showing adversarial training will effectively regularize the non-robust component. Also, based on our theory, we point out the possible direction of achieving simultaneous adversarial robust and decent standard performance, which is enabling the model to rely more on robust component to distinguish data from different categories.
>
> All these points presented above are actually the central point of our work, and we believe we have stated clearly in our main text.
>
> Also, we think most of our mathematical derivations are rigorous. The example you give is only meant to serve as an alternative intuition, and is therefore put in the appendix. It is not in the main flow of our work.

---

### Author Response · Authors · 2019-11-11
**Restating the Major Contribution of This Submission**

It seems that there are some serious misunderstanding in our submission among reviewers. Therefore, we have to restate our major contribution here again: in this work, we theoretically reveal the relation between the standard performance and adversarial robustness by disentangling the performance metric. Additional experimental results are provided to help prove our theoretic conclusions that there are non-robust component and robust component in the performance metric. We're disappointed that despite we stressed this contribution multiple times in the paper, it is not noticed by some reviewers.

---

### Decision · Program_Chairs · 2019-12-19

**Decision:**

Reject

**Comment:**

All reviewers suggest rejection. Beyond that, the more knowledgable two have consistent questions about the motivation for using the CCKL objective. As such, the exposition of this paper, and justification of the work could use improvement, so that experienced reviewers understand the contributions of the paper.